# DECOUPLING DYNAMICS AND REWARD FOR TRANSFER LEARNING

**Amy Zhang**,* **Harsh Satija**,* **Joelle Pineau**
Department of Computer Science, McGill University
Facebook AI Research
{amyzhang, hsatija}@fb.com, jpineau@cs.mcgill.ca

## ABSTRACT

Current reinforcement learning (RL) methods can successfully learn single tasks, but often generalize poorly to modest perturbations in task domain or training procedure. In this work we present a decoupled learning strategy for RL that creates a shared representation space where knowledge can be robustly transferred. We separate learning the task representation, the forward dynamics, the inverse dynamics and the reward function of the domain, and show that this decoupling improves performance within task and transfers well to changes in dynamics and reward.

## 1 INTRODUCTION

Reinforcement Learning (RL) provides a sound decision-theoretic framework to optimize the behavior of learning agents in an interactive setting. However, one of the limitations to applications of RL to real-world tasks is the amount of data required for learning an optimal policy. Our goal is to design an RL model that can be efficiently trained on new tasks, and produce solutions that generalize well beyond the training environment. We take inspiration from Successor Features (Dayan, 1993), which decouples the value function representation into dynamics and rewards, and learns them separately. We take this further by explicitly decoupling learning the state representation, reward function, forward dynamics, and inverse dynamics of the environment. We posit that we can learn a representation space $\mathcal{Z}$ via this decoupling that makes downstream learning easier as: (1) the modules can be learned separately enabling efficient reuse of common knowledge across tasks to quickly adapt to new tasks; (2) the modules can be optimized jointly leading to a representation space that is adapted to the policy and value function, rather than only the observation space; (3) the dynamics model enables forward search and planning, in the usual model-based RL way. Our approach is the first model-based RL method to explicitly incorporate learning of inverse dynamics, and we show that this plays an important role in stabilizing learning.

## 2 DECOUPLING MODEL-BASED RL

Consider an RL agent deployed in an environment which is modelled as a Markov Decision Process (MDP), i.e., defined by a set of states $\mathcal{S}$, a set of actions $\mathcal{A}$, dynamics $p(\cdot|s, a)$, and rewards $r(s, a)$. The behavior of the RL agent is defined by a policy $\pi : \mathcal{S} \rightarrow \mathcal{A}$, specifying an action to apply in each state. The goal is to learn an optimal policy, denoted $\pi^*$, that maximizes the expected cumulative reward over trajectories. The value function $V^\pi(s)$ and state-action value function $Q^\pi(s, a)$ are defined as usual in the RL literature (Sutton & Barto, 1998). We define $\mathcal{T}$ to be the space of tasks that share $\mathcal{S}$ and $\mathcal{A}$, but dynamics $p(\cdot|s, a)$, and rewards $r(s, a)$ can vary. We sample from $\mathcal{T}$ at training time. When the agent is in a particular task $\mathcal{T}_k$, it collects a set of trajectories, $D^{\mathcal{T}_k} = \{D_1^k, D_2^k, ..., D_n^k\}$, where $D_i^k = \{s_0, a_0, s_1, a_1, \ldots, s_t, a_t, \ldots, s_{T-1}, a_{T-1}, s_T\}$.

Our objective is to provide a modular framework for model-based RL, leveraging a decomposition of the learning problem to provide reusable components that can be bootstrapped to enable fast re-training following changes in dynamics and rewards. The learning is decomposed into two

---

*Equal contribution.

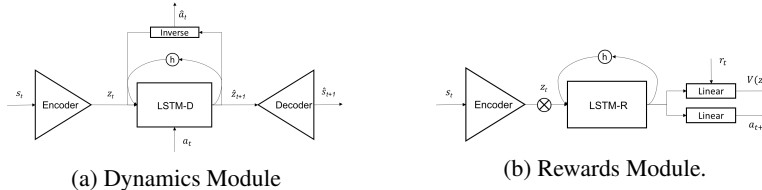

(a) Dynamics Module        (b) Rewards Module.

Figure 1: $\otimes$ denotes the stop gradient operator, which doesn't allow the gradients to propagate back.

complementary objectives, one for learning the state dynamics model and the other for learning the reward function. Figures 1a & 1b give an overview of the proposed architecture.

**A Modular Dynamics Model.** The goal of this module is to learn the dynamics of the environment $p(\cdot|s,a)$. We have an encoder and decoder pair, $f_{enc}(s;\theta_{enc})$ and $f_{dec}(z;\theta_{dec})$, that allows us to learn a mapping between state space $\mathcal{S}$ and representation space $\mathcal{Z}$. We define a forward model, $f_{for}(s,a;\theta_{for})$, which predicts the transition probability $p(\cdot|s,a)$ in the $\mathcal{Z}$ space. The forward model is learned using a recurrent architecture so the latent representation can incorporate temporal dependencies, where $h_t$ denotes the hidden state of the recurrent model[1]. We add an additional inverse model $f_{inv}(s,s';\theta_{inv})$, which observes the current and next state $z,z' \in \mathcal{Z}$, and aims to predict which action was taken. We also posit the inverse model is a necessary constraint to maintain causality in $\mathcal{Z}$. As we learn the forward and inverse models in $\mathcal{Z}$, we define the four functions:

$$z_t = f_{enc}(s_t;\theta_{enc}), \qquad\qquad \hat{z}_{t+1}, h_t = f_{for}(z_t, a_t, h_{t-1};\theta_{for}),$$
$$\hat{s}_t = f_{dec}(z_t;\theta_{dec}). \qquad\qquad \hat{a}_t = f_{inv}(z_t, z_{t+1};\theta_{inv}).$$

We first define the total dynamics loss:

$$\mathcal{L}_{dynamics}(\theta_{dynamics}) = \sum_{t=0}^{T}(\lambda_{dec}\mathcal{L}_{dec} + \lambda_{for}\mathcal{L}_{for} + \lambda_{inv}\mathcal{L}_{inv})$$

where $\lambda_{dec}$, $\lambda_{for}$, $\lambda_{inv}$ are constant hyper-parameters. The decoder loss, $\mathcal{L}_{dec}$, consists of the reconstruction loss between $s_t$ and $\hat{s}_t$ and the next state prediction loss between $\hat{s}_{t+1}$ and $s_{t+1}$.

$$\mathcal{L}_{dec}(\theta_{enc}, \theta_{dec}, \theta_{for}) = (\hat{s}_t - s_t)^2 + (\hat{s}_{t+1} - s_{t+1})^2.$$

The forward model loss and inverse model losses are similarly defined as:

$$\mathcal{L}_{for}(\theta_{for}, \theta_{enc}) = (\hat{z}_{t+1} - z_{t+1})^2, \qquad\qquad \mathcal{L}_{inv}(\theta_{inv}) = (\hat{a}_t - a_t)^2.$$

Where $\hat{a}_t \sim p(\hat{a}) = f_{inv}(z_t, z_{t+1};\theta_{inv})$.

By abstracting away the model of dynamics to a representation space $\mathcal{Z}$, we can encode more or less information than what exists in the given space $\mathcal{S}$. We show that this abstraction allows for easier learning and improved results across a variety of environments. Note that this module learns a dynamics model purely with respect to trajectories; it ignores tasks and rewards.

**A Modular Reward Model.** The goal of the Reward Module is to learn the value function and policy over $\mathcal{Z}$ instead of $\mathcal{S}$. The reward module is the primary decision-making module – it selects the next action and predicts the expected value. We use an Actor-Critic method (Sutton et al., 2000) to learn the policy and value function simultaneously:

$$\pi(a_t|z_t;\theta_{actor}) = f_{actor}(z_t;\theta_{actor})$$
$$V(z_t;\theta_{critic}) = f_{critic}(z_t;\theta_{critic})$$

using TD learning with multi-step bootstraps (Sutton, 1988).

**Training the Decoupled Model.** We train the dynamics module in a supervised manner and off-policy on trajectories generated with actions randomly drawn from a uniform distribution, since it is decoupled from any specific task. We can also bootstrap data collected from previous tasks, and exhibit more stable learning compared to on-policy training. Assuming that the policy used to collect the data is sufficiently exploratory, we are able to learn a representation space that captures useful information for a family of tasks. Clearly there is a trade-off here: more exploration provides more robust information, but is less efficient than a narrowly targeted policy.

---

[1]We use an LSTM (Hochreiter & Schmidhuber, 1997) as the recurrent model

The representation space (encoder) is static while we train the reward module. We first train the full dynamics module with sample trajectories collected offline (either off- or on-policy) then freeze the encoder weights before training the reward module online from this fixed encoder. The rewards module is trained online and on-policy, using an actor-critic approach analogous to A3C (Mnih et al., 2016), with the distinction that the actor and critic operate on the representation space $\mathcal{Z}$ built by the dynamics module.

## 3 EXPERIMENTS, ABLATIONS, AND RESULTS

We first consider the Simple Generalization case (Table 1[2]). A3C (Mnih et al., 2016) is the most suitable baseline here due to the efficiency it achieves through parallelization. The main trunk of our architecture is the same for our method and the baseline for fair comparison. We evaluate on continuous control tasks in MuJoCo (Todorov et al., 2012) and find that our model-based approach significantly outperforms standard A3C.

Next we evaluate dynamics and reward transfer. For reward transfer, we modify the task by negating the reward given by the environment[3]. In Table 2, for reward transfer – the more negative the score, the better. For dynamics transfer, we increase density and damping on the joints. Finally, we consider the case where both the reward and dynamics change. In all the transfer scenarios considered, the results in Table 2 show a consistent advantage for DDR (Decoupled Dynamics and Reward, our method) which is able to leverage pre-trained modules for the components that do not change.

| MODEL | SWIMMER | ANT | HOPPER | HALFCHEETAH |
|-------|---------|-----|--------|-------------|
| A3C   | 55.4    | 24.3| 8      | 124.8       |
| DDR   | **68**  | **508** | **36** | **869**  |

Table 1: Main Results. Reward averaged over 5 runs, evaluated over 100 trajectories, trained for 1M episodes.

| TASK | MODEL | CHANGE IN REWARD | CHANGE IN DYNAMICS | CHANGE IN BOTH |
|------|-------|------------------|--------------------|-----------------|
|         | DDR     | **-86.3** | **66.9** | **-65** |
| SWIMMER | A3C (F) | 0.6       | 50.9     | -5.1    |
|         | DDR     | **-908**  | **793**  | **-366** |
| ANT     | A3C (F) | -11.8     | 50       | -50.8   |

Table 2: MuJoCo Transfer Experiments. A3C (f) = A3C finetuned. Reward transfer in this case is negating the reward, so the more negative the better.

We next selectively ablate components of the dynamics model. Table 3 shows a significant drop in performance when removing the forward model, but more surprisingly, an even greater drop in performance when removing the inverse model. This suggests that the inverse model is essential for regularizing the dynamics problem in preventing degenerate solutions; an important finding of this work. Using a plain auto-encoder performs even worse. These results confirm that learning dynamics with all four components is crucial for a good representation space.

| AGENT | FULL | NO F | NO I | AE |
|-------|------|------|------|-----|
| SWIMMER | **68**  | 25.9 | 4.48 | -3.3 |
| ANT     | **508** | 281  | 80.5 | 37.5 |

Table 3: Ablation results averaged over 5 runs. Full = All four losses, No F = No forward model, No I = No inverse model, AE = Autoencoder (no forward or inverse models).

In conclusion, we present a decoupled model-based RL framework that offers efficient and modular reuse to pre-trained models and enables robust transfer across tasks. By learning an encoder jointly with the dynamics we can focus representation on relevant information. The incorporation of an inverse model has an important stabilizing effect on the dynamics model. The modularity of the rewards model allows off- and on-policy learning as well as could be extended to other policy optimization methods such as TRPO (Schulman et al., 2015) or PPO (Schulman et al., 2017).

---

[2]We set a maximum episode length of 500 for evaluation. Other work does not specify the episode length used for the same environments, so our results are not directly comparable.

[3]The reward is computed as reward = forward_reward - ctrl_cost - contact_cost + survive_reward. Maximizing the negative reward is not so simple as merely maximizing negative velocity.

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
