# OpenReview forum: "Decoupling Dynamics and Reward for Transfer Learning"
_ICLR.cc/2018/Workshop — Accept_

### Official Review · AnonReviewer1 · 2018-02-21
**interesting topic, hard to follow**

**Rating:** 7
**Confidence:** 4

**Review:**

The terse workshop submission format makes it really hard to understand what's going on here. Briefly, it appears that one aspect of the work is to learn a model so that model-based decision making can be carried out. (How is decision making done? Is that a matter of using an LQR approach or something?) Another aspect of the work is to learn an encoding (abstraction?) of state so that the model (and planning?) can be made more efficient. Another aspect of the work is transfer---using learning from one environment to improve learning/performance/planning in another.

The most interesting possible takeaway message for me is the idea that an inverse model (guess the action given the state and next state) plays an essential role in making the learning work. I would love to have seen the paper just focus on this one aspect instead of covering transfer and other topics leaving only room for ablation results that show that turning this aspect off hurts performance. It strikes me that there's a lot more to the story.

---

### Official Review · AnonReviewer4 · 2018-03-04
**Interesting approach but presentation is lacking in terms of justification and experimental validation**

**Rating:** 5
**Confidence:** 4

**Review:**

This paper presents a framework for model-based reinforcement learning in which many of the model components are decoupled. In particular, the authors propose to learn the dynamics of the environment using an encoder model, a decoder model, a forward model, and an inverse model, and they propose to learn a policy and value function in a latent representation space instead of the original state space, building upon the work of [Dayan, 1993].

I found the paper to be generally clear and well-written, and the proposed framework seems sufficiently novel and broad in scope to be an interesting contribution to the model-based reinforcement learning community. The ablation analysis is also helpful for providing better clarity on the incremental value of each component of their proposed approach.

At the same time, I did have a few questions/concerns about this work:
1) It's not clear to me why LSTMs are needed for the encoder-decoder components of the dynamics model. Since the dynamics are given by an MDP, all of the relevant information should be captured in the current state.
2) The authors do not motivate why learning the reward model in the latent space is better than in the original space.
3) It still isn't clear to me how much of the performance improvement is due to learning a better dynamics model (using the decoupled framework) and how much is due to finding a latent variable space in which it is easier to learn a good policy.

Other specific comments and questions:
1) The LSTM architecture used in the experiments is not specified.
2) The dynamics loss hyperparameters are not specified in the experiments. How sensitive are the results to this choice? Was the tuning of these values performed within the 1M episodes?
3) There are no error bars for the experiments.
4) The authors do not specify whether the 1M episodes used in the experiments includes the data used to train the dynamics module. If it did include the data used for training the dynamics model, then the authors do not explain how the data was split. If it did not include this data, then the experimental comparison seems unfair.
5) I think the authors should compare their approach against at least one other model-based approach. Comparing against a single model-free algorithm seems insufficient.
6) The authors claim that incorporating the inverse model has "an important stabilizing effect on the dynamics model", but I don't see how this was shown.

I think the paper rates reasonably well in terms of clarity and novelty, but that there is room for improvement in terms of significance and quality.

---

### Official Review · AnonReviewer2 · 2018-03-09
**Important paper on model-based RL by decoupling dynamics from reward models**

**Rating:** 8
**Confidence:** 4

**Review:**

In this paper on deep reinforcement learning, the authors address multi-task learning and transfer learning to a new task in model-free RL by decoupling 1) learning of a dynamical model for the states and actions from 2) learning a task-dependent reward model, effectively transforming the problem to model-based RL where a forward model of the dynamics (state transitions from s_t to s_{t+1} in a representation space Z) and an inverse model of the dynamics (going from s_t and s_{t+1}, through the representation space Z, to the action a_t that cause the change of state) are explicitly learned. This is achieved by using an encoder/decoder architecture with a dynamics LSTM and an inverse decoder in order to learn the state representation, then by freezing the encoder and representation and training policy LSTMs on the frozen representations of the state.

The idea is very good, well formulated and well executed.

I have some questions though, after reading the paper:
This approach also outperforms A3C on several MuJoCo tasks (swimmer, ant, hopper and half-cheetah) on multi-task learning (table 1). Can the authors remind the reader if the input is visual or joint angles? The authors use the words "generalization" and "transfer", which is confusing; for table 2, is only the encoder frozen and is the policy LSTM retrained on each new task, or is the policy LSTM frozen? Is training the dynamics module on random actions scalable to high-dimensional problems (in terms of actions) and to problems with long-range dependencies?

---

### Public Comment · (anonymous) · 2018-05-11
**Details about the experiments**

The paper is really interesting, thank you.

Would it be possible to obtain the training details of the Dynamics and Reward modules ?

The best would be to open source the code. If it's not possible, could you please give pseudo-code details so that I can try to reproduce your results ? I would need all hyperparameters and training procedure (you might have used some tricks that A3C use to facilitate training).

---

> ### Author Response · Authors · 2018-05-11
> **Opensourcing code**
>
> Thank you for your interest! We are planning to release the code as soon as possible after the NIPS deadline. It just needs some cleanup. We will post here when the code is publicly available, and update the paper on arxiv with a link.

---

### Decision · Program_Chairs · 2018-03-20
**ICLR 2018 Workshop Acceptance Decision**

**Decision:**

Accept

**Comment:**

Congratulations, your paper was accepted to the ICLR workshop.